# Gaps between Open Science activities and actual recognition systems: Insights from an international survey

Florencia Grattarola[1], Hanna Shmagun[2]*, Christopher Erdmann[3], Anne Cambon-Thomsen[4,5], Mogens Thomsen[4], Jaesoo Kim[6], Laurence Mabile[4]

1 Faculty of Environmental Sciences, Czech University of Life Sciences Prague, Prague, Czech Republic, 2 Korea Institute of Science and Technology Information, Seoul, South Korea, 3 SciLifeLab, Uppsala, Sweden, 4 CERPOP, INSERM and Université de Toulouse III Paul Sabatier, Toulouse, France, 5 CNRS, Toulouse, France, 6 Hongik University, Seoul, South Korea

☯ These authors contributed equally to this work.
* hanna.shmagun@gmail.com

**Data Availability Statement:** All relevant data are within the manuscript and its Supporting Information files.

## Abstract

There are global movements aiming to promote reform of the traditional research evaluation and reward systems. However, a comprehensive picture of the existing best practices and efforts across various institutions to integrate Open Science into these frameworks remains underdeveloped and not fully known. The aim of this study was to identify perceptions and expectations of various research communities worldwide regarding how Open Science activities are (or should be) formally recognised and rewarded. To achieve this, a global survey was conducted in the framework of the Research Data Alliance, recruiting 230 participants from five continents and 37 countries. Despite most participants reporting that their organisation had one form or another of formal Open Science policies, the majority indicated that their organisation lacks any initiative or tool that provides specific credits or rewards for Open Science activities. However, researchers from France, the United States, the Netherlands and Finland affirmed having such mechanisms in place. The study found that, among various Open Science activities, Open or FAIR data management and sharing stood out as especially deserving of explicit recognition and credit. Open Science indicators in research evaluation and/or career progression processes emerged as the most preferred type of reward.

## Introduction

Open Science (OS) has emerged as a transformative paradigm in the domain of scientific research. Fundamentally, OS underlines the value of transparency as a cornerstone of scientific activities and emphasises the sharing of research data, methods and outputs with the broader scientific community and the general public. Making scientific knowledge openly available enables the replication, verification and validation of research findings, thereby fostering greater scientific rigour and reproducibility. OS also encourages interdisciplinary

**Funding:** This research received funding from the PARSEC Belmont Forum Collaborative Research Action on Science-Driven e-Infrastructures Innovation (SEI2018). Additionally, FG was funded by the European Union (ERC, BEAST, 101044740). HS received funding from the Korea Institute of Science and Technology Information (No. K-24-L01-C05-S01). ACT received funding from the European Union's Horizon 2020 research and innovation programme EOSC Future under grant agreement no 101017536 as domain ambassador. The funders had no role in study design, data collection and analysis, decision to publish, or preparation of the manuscript.

**Competing interests:** The authors have declared that no competing interests exist.

collaboration, enabling researchers from diverse fields to work together, thereby accelerating scientific progress and facilitating breakthrough discoveries [1].

This new paradigm represents a shift towards a more inclusive approach to science, where knowledge is collectively built and disseminated, ultimately benefiting society as a whole and fostering public trust in science. Despite the numerous potential benefits of OS [2], and various initiatives to facilitate sharing activities [3], they have not yet become the norm due to various factors hindering their widespread adoption, as also underlined in the report of the European project ON-MERRIT [4]. The traditional research evaluation and reward system is seen as one of the most significant inhibitors of OS [5–8]. This system predominantly relies on quantitative metrics such as impact factor, citation counts and the number of publications [9]. Within this framework, OS practices like data and code sharing often remain overlooked and not adequately rewarded, nor are they typically included in performance indicators for promotion and tenure. For instance, a recent survey involving researchers who have served on grant review, hiring or promotion committees confirmed that these committees still mainly use proxies such as journal reputation and impact factor, while the transparency of research outputs and their open sharing, integral to OS practices, are among the least used evaluation criteria. Nevertheless, the surveyed researchers exhibited dissatisfaction with judging credibility using these traditional proxies and were receptive to new solutions [10]. Likewise, Pontika et al. [4] showed that criteria relating to OS practices are essentially not considered important in current research evaluation for promotion decisions at higher education institutions, whereas securing research funding and publishing in prestigious journals or conferences are prioritised.

Although there are emerging global movements aiming to promote reform of the traditional research evaluation and reward systems, such as the Coalition for Advancing Research Assessment (https://coara.eu/) and the Evaluation of Research Interest Group (https://rd-alliance.org/groups/evaluation-research-ig), a comprehensive big picture of the existing best practices and efforts across various institutions to integrate OS into these frameworks remains underdeveloped. It is also important to explore researchers' perceptions towards rewarding their sharing activities and efforts to practise OS to ensure a better endorsement of the rewarding mechanisms, yet there is a notable scarcity of research in this particular area. Studies such as [11,12] are some of the few that partially addressed the types of rewards for sharing of intermediate resources such as research data. To address these gaps, we undertook a survey to identify perceptions and expectations of various research communities around the world regarding how OS activities are (or should be) formally recognised and rewarded. This survey was developed as part of SHARC's work (SHAring Reward & Credit), an interdisciplinary group established under the framework of the Research Data Alliance (RDA) to investigate and promote crediting and rewarding mechanisms for OS activities.

## Methods

We generated an online anonymous survey using the LimeSurvey software. The survey consisted of 19 questions, including yes or no responses, multiple-choice options, Likert rating scales and open-ended questions (see the questionnaire in the S1 File). Prior to distribution, the questionnaire was tested in each language by the authors' colleagues to ensure clarity and comprehensibility of the questions. The survey was available in English, Korean and Spanish and distributed using a snowball sampling approach by email, social media (e.g., Twitter and Slack) across the authors' professional networks (e.g., life sciences, geophysical, and medical communities and information service providers) and the RDA community. See the full list in the S2 File. It was run between 23 May and 30 September 2022.

This survey-based study received approval number 2022–507 on the 20th of May 2022 by the Research Ethics Committee of the University of Toulouse. All respondents provided written informed consent for anonymous participation and the processing of their responses. They expressed their voluntary opt-in, after they had read information about the research project (including information on data processing), by clicking the 'agree' button on the online survey platform. Participation in the survey was only possible after this step. Additional information regarding the ethical, cultural and scientific considerations specific to inclusivity in global research is included in the S3 File (Checklist).

The survey consisted of 5 sections that aimed to assess (1) respondents' profiles, (2) respondents' familiarity and engagement with OS, FAIR (Findable, Accessible, Interoperable, Reusable) Principles and awareness of related institutional policies, (3) respondents' preferences on which OS activities (see Table 1) should be rewarded, (4) current rewarding initiatives or tools and (5) how respondents would want to be rewarded.

The free text provided by respondents regarding their disciplines (they were asked to choose up to three main disciplinary fields) was classified into a unique field and subfield according to OECD [14] fields of research and development. One respondent did not indicate any discipline, while three others provided insufficient information to correctly classify their disciplinary field (e.g., 'Multidisciplinary' and 'Science').

Open-ended responses for assessing how researchers prefer to be rewarded were categorised according to the terminology derived from a mixed coding approach [15]. First, we built a terminology framework to create a common understanding of rewarding and recognition elements between investigators, applying concept-driven coding. Then, we used data-driven coding based on survey responses, i.e., we used this terminology framework and extended it to include new categories that came out from collected open-ended survey responses. Each response could have more than one preferred type of reward. To represent graphically the researchers' preferences, we counted the frequency of a term being mentioned over the total number of respondents. For example, if the term 'funding/grants for OS activities' was mentioned by 10 respondents, it would mean 4.3% of them preferred this type of reward.

The submitted responses were processed in R software [16], relying on the 'tidyverse' package [17]. Likert figures were produced using the 'likert' package [18], while the disciplines figure was created using 'webr' [19]. To assess the reliability of the questions that used binary, Likert and ordinal scales, Cronbach's alpha was calculated, revealing an acceptable internal consistency with a value of 0.71 (CI = 0.63, 0.76). To see the code for all the analyses, including the data, access Grattarola [20] or our GitHub repository at https://github.com/bienflorencia/rda-sharc-survey.

The preliminary results of the survey were presented and discussed during the International Data Week 2022 (https://www.rd-alliance.org/plenaries/rda-20th-plenary-meeting-gothenburg-hybrid/towards-implementable-recommendations-taking-0).

**Table 1. Open Science activities considered in the survey.** Adapted from [13]). Respondents had the possibility to indicate other types of OS activities in an open-ended question.

| OS activities considered in the survey |
| --- |
| Sharing a research manuscript as a preprint |
| Publishing a paper or monograph book as open access |
| Preregistration of the study design, methods, hypothesis etc., prior to commencing the research |
| Open or FAIR data management and sharing (for research data, software, models, algorithms, workflows etc.) |
| Participation in open peer review (being reviewed or the reviewer) |
| Participation in public engagement, including citizen or community science |
| Collaboration via virtual research environments or virtual laboratories |

## Results

### Respondents' profiles

We received 230 responses, with a survey's completion rate of over 62%, from individuals across five continents covering 37 countries: 21.3% from France, 18.3% from the United States, 17.8% from South Korea, 5.2% from Uruguay, 4.8% from Argentina, 3.9% from Germany, 3.5% from United Kingdom, 3% from the Netherlands, 2.2% from Brazil, and 20% from other countries. 41.3% declared their gender as female, 35.7% as male, 1.3% as non-binary or gender-queer, and 21.7% did not provide a response. Most respondents had either 'Researcher' (40.9%) or 'Professor' (15.2%) as their primary job title and had between 10 to 20 years of experience in their field (32.2%). They were affiliated with a 'University' (39.1%), a 'Research institute' (37.4%) or a 'Government agency' (12.6%). Fig 1 maps the respondents' main disciplines. The two main disciplines are 'Natural sciences' (53.1%) and 'Social sciences' (22.6%). Aggregated responses can be found in the S4 File (note that open-ended responses are excluded because they contain details that could be cross-referenced with other data sources, potentially leading to deanonymisation).

### Familiarity and engagement with OS & FAIR principles and awareness of related institutional policies

Ninety per cent of the respondents claimed to be familiar with OS as a concept. From the range of proposed OS activities, most respondents (63%) were predominantly involved in

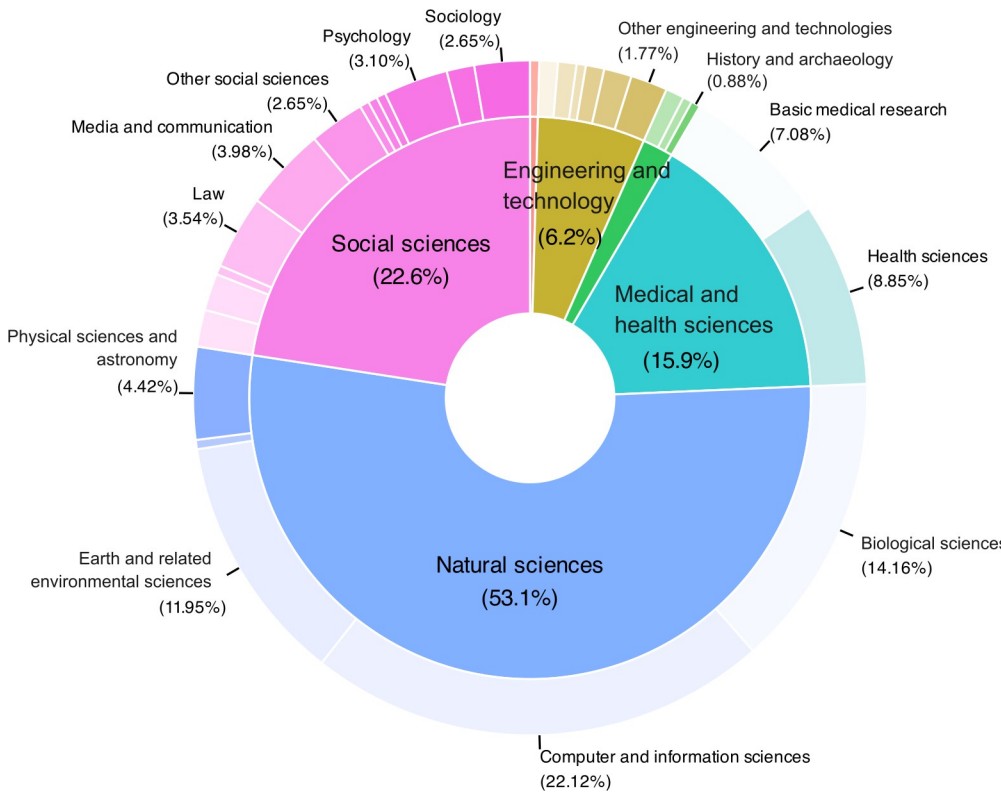

**Fig 1. Distribution of the respondents' disciplines.** Main fields are shown inside, and subfields outside. Fields are according to the OECD [14]. The percentages shown for both nested levels are calculated based on the total number of respondents minus 4 responses that were not possible to categorise.

making their scientific publications open access, while only a few of them (16%) reported experience with pre-registering their study designs (Fig 2). In an open-ended question, respondents also mentioned other OS activities they were involved in, including OS teaching/training, promoting or supporting the development of OS in their institutions, managing OS communities, building best practices and policies for implementing OS activities and maintaining repositories and other digital infrastructures.

Regarding the FAIR Principles, 74% of the participants responded to be familiar with them, while 26% were not. Almost one-third (32.2%) claimed to be involved in some steps of the data FAIRification process [21], while 40.9% said not to be involved, and 27% did not answer. Of those involved, around one quarter mentioned being involved in all steps of the data FAIRification process (26.3%).

More than half of the respondents (55%) replied that their organisation had formal policies on OS activities, while 43% said they did not have any, and 3% did not answer. The mentioned policies were mostly focused on open research outputs, such as publications, research data (e.g., data management plans) and software. Some examples of institutional OS policies, for instance, included the Open Access policy of the Korea Institute of Science and Technology Information/KISTI (South Korea), Research data management strategy of Research Centres of Catalonia/CERCA (Spain), Open Access policy from the National Agency of Research and Innovation/ANII (Uruguay), Research data policy of the National Institute of Geophysics and Volcanology/INGV (Italy) and the CNRS Roadmap for Open Science (France). The latter was the only reported example of policies that include specific OS incentives, particularly in research evaluation.

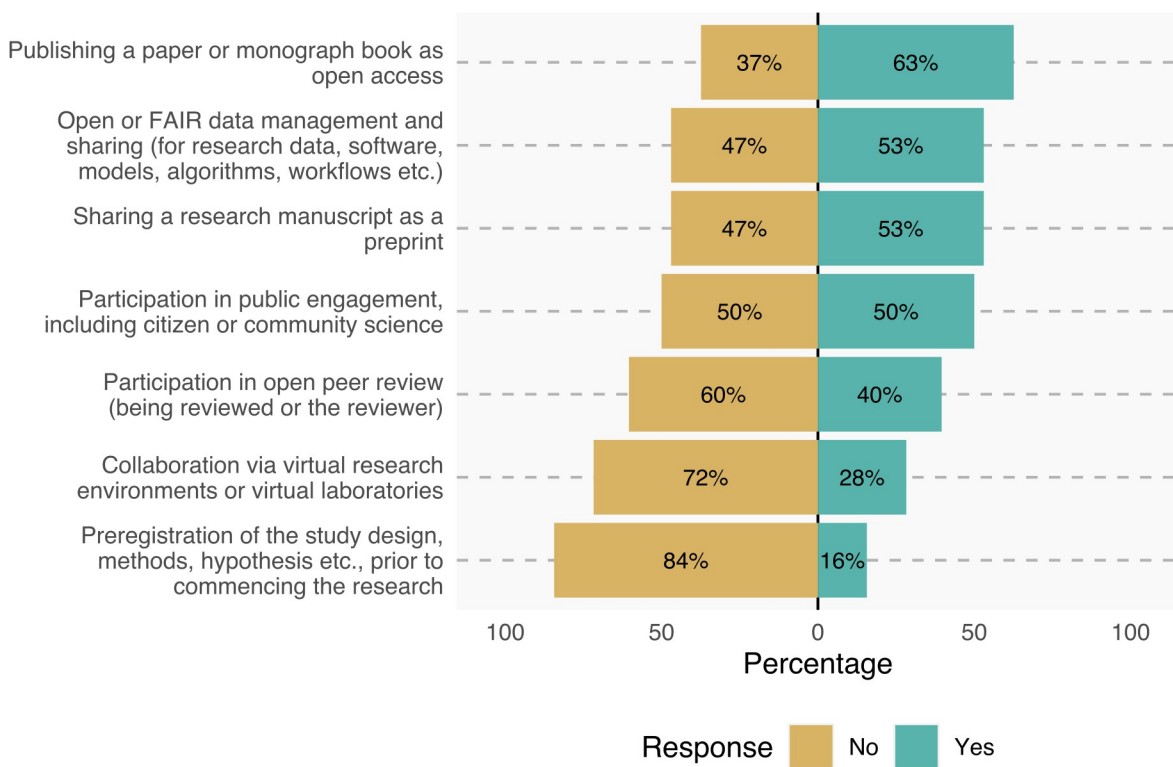

**Fig 2. Responses to the question, 'Are you involved in some of the following Open Science activities?'.** Text on the left is Open Science activities proposed in the survey.

## Researchers' preferences on which OS activities should be rewarded

Participants were asked to indicate their level of agreement or disagreement regarding whether proposed OS activities should be rewarded (Fig 3). In the questionnaire, we mentioned that rewards could include, for example, career promotion, grants/funding/prizes, gained credits in a research evaluation procedure, authorship/ contributorship and increased academic visibility. Open or FAIR data management and sharing was the most endorsed OS activity (82% mentioned that this activity should be rewarded, either 'definitely' or 'very probably'), followed by Publishing a paper/monograph as open access (79%). In contrast, the least endorsed activity was Sharing a research manuscript as a preprint, with 14% of the participants agreeing that this should 'probably not' or 'definitely not' be rewarded. Over half of the respondents were neutral (51.5%) on whether collaboration via virtual research environments needs specific rewards or not.

## Current rewarding initiatives or tools

Most of the participants (85%) replied that their organisation does not have any initiative or tool which gives credits/rewards for OS activities. Those who replied positively were mainly people working in France, the United States, the Netherlands, Finland, Slovenia, Spain, the United Kingdom and Germany. When asked to point to examples of initiatives or tools, respondents mentioned specific funds, monetary prizes, awards, badges and OS activities

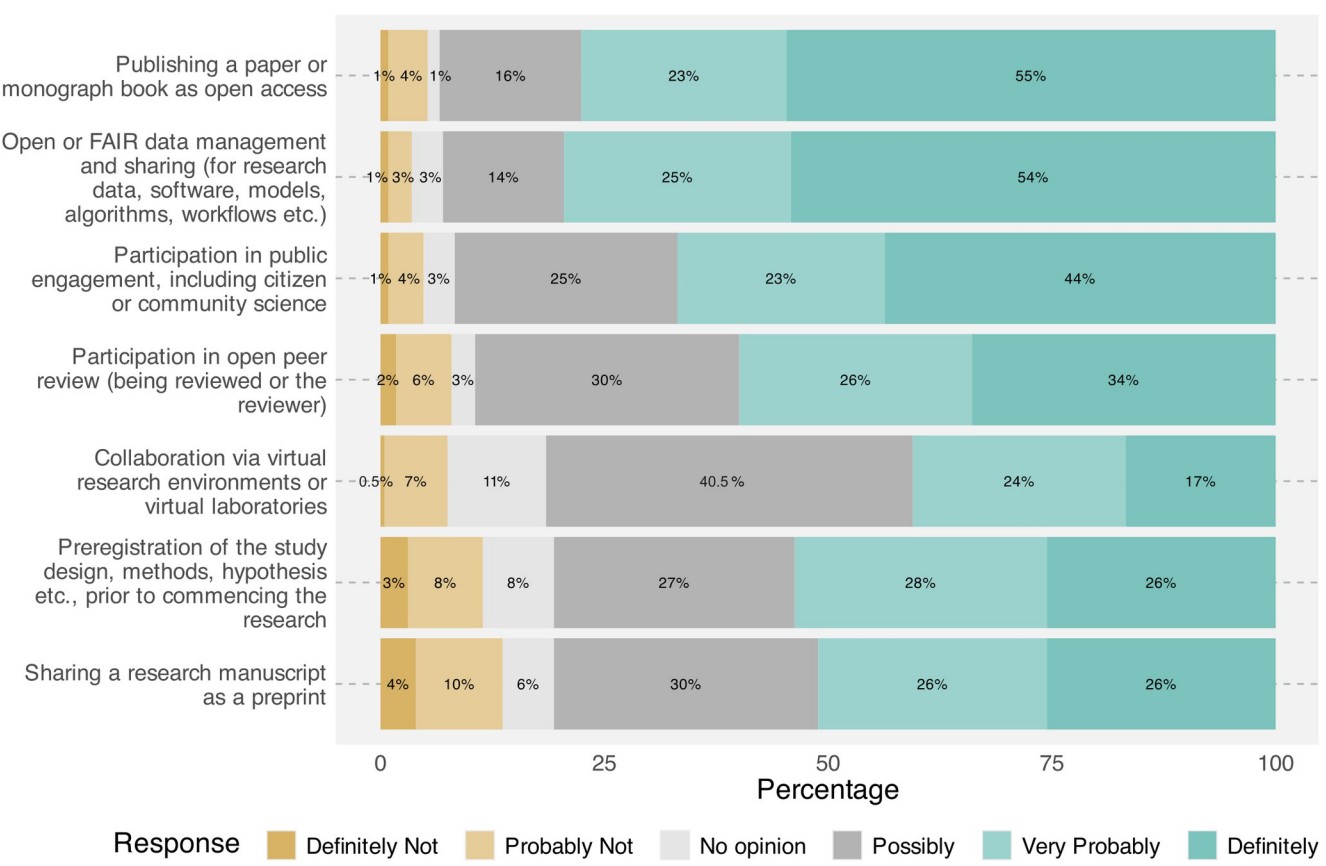

**Fig 3. Responses to the question, 'Could you please specify to what extent you feel the following activities should be credited/rewarded?'.** Text on the left is Open Science activities proposed in the survey.

being considered in the researchers' evaluations. For instance, for researchers of the French National Centre for Scientific Research (French: Centre national de la recherche scientifique, CNRS), only publications available in the national open repository (HAL) are eligible to be reported in the researcher's annual activity report; other publications are not considered as research outputs in the researcher performance evaluation. Another example includes the Open Science Recognition Prize (https://www.agu.org/honors/open-science) of the American Geophysical Union (AGU), where three awards are given annually for recognising work in advancing OS in Earth and Space Science.

## Which types of rewards are most preferred by researchers

As mentioned in the Methods section, we applied a mixed coding approach to develop a terminology for credits and rewards (see the final terminology framework in Table 2). On average, each respondent suggested 1.8 types of rewards. While 176 people (76.5%) responded to this question, 54 (23.5%) did not mention any rewards. The most desired rewards for OS activities are presented in Fig 4. According to the results, the most preferred type of reward was OS indicators in research evaluation and/or career progression processes, with 54.5% of respondents favouring it. This was followed by funding or grants for OS activities, which 21.3% preferred, and specific OS awards/bonuses, chosen by 14.3%.

## Discussion and conclusions

In summarising our findings, the vast majority of respondents are familiar with the concept of OS in general terms, predominantly engaging in making their scientific publications open access. This supports the findings of previous survey studies, which concluded that open access

**Table 2. Terminology for Open Science credits and rewards.**

| Category | Term |
|---|---|
| policies | OS indicators in research evaluation and/or career progression processes (e.g., considering open access publications and high-quality FAIR datasets when making decisions for research evaluation, promotion and tenure) |
| tangible rewards | funding/grants for OS activities |
| tangible rewards | awards/bonuses |
| tangible rewards | research visibility indicators |
| tangible rewards | authorship/contributorship |
| capacity building or support | capacity building for OS (e.g., training, raising awareness, provision of IT tools) |
| tangible rewards | acknowledgement/citation |
| policies | support through regulations and policy mandates |
| tangible rewards | collaboration (e.g., joint research, co-authorship) |
| intangible rewards | contribution to 'good' science, research quality and integrity |
| capacity building or support | OS certifications/badges |
| intangible rewards | science as a public good |
| tangible rewards | CRediT taxonomy |
| tangible rewards | financial contribution to reviewers of open access journals |
| tangible rewards | OS activities included in working hours |
| sanctions or penalties | punishment for 'closed' science |
| intangible rewards | research reputation |
| capacity building or support | championships/contests |

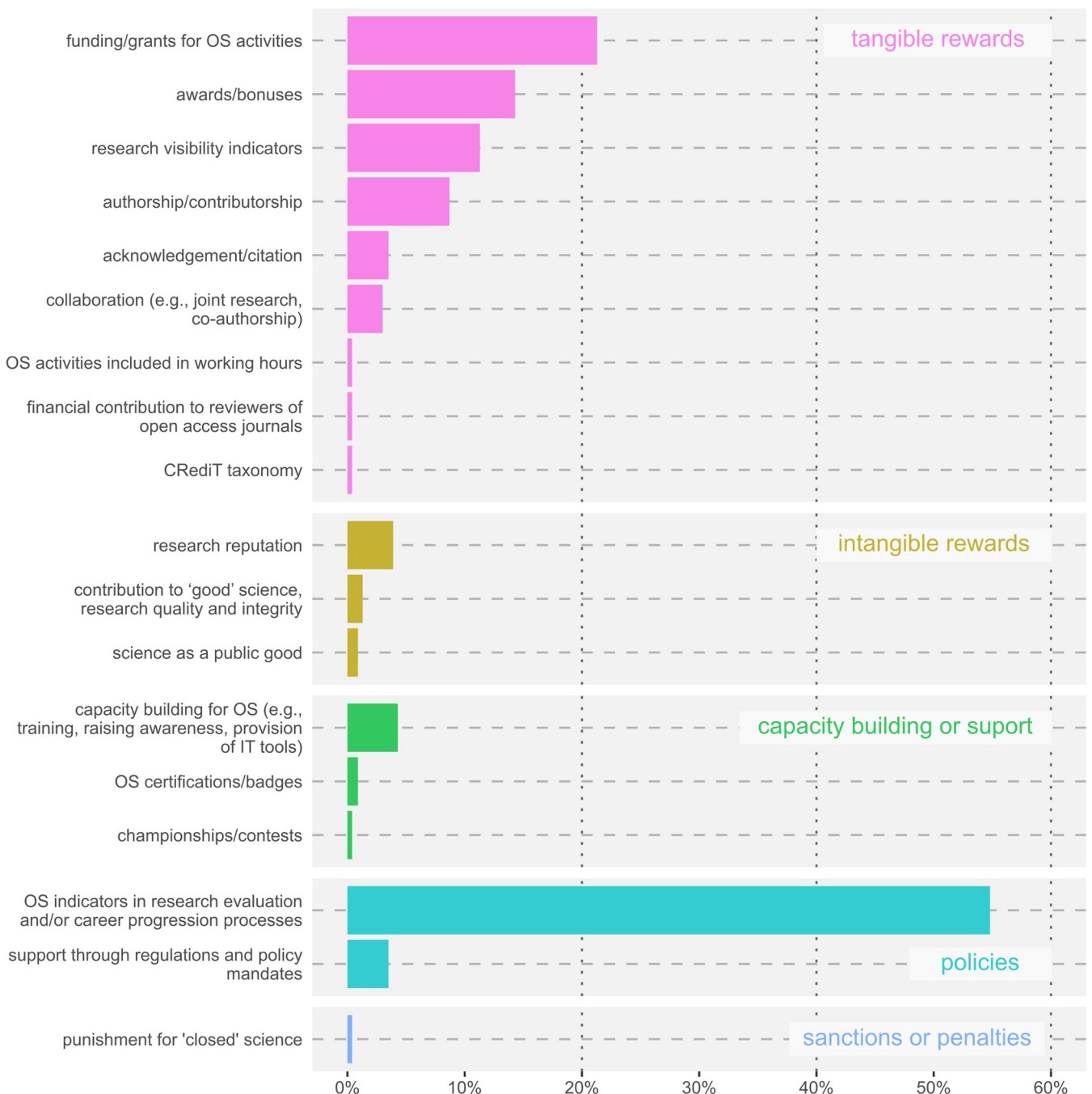

**Fig 4. Responses to the open-ended question, 'How would you want the previously mentioned Open Science activities to be rewarded?', categorised according to this study's proposed terminology.** Text on the left is the types of rewards categorised, while on the right the main category associated with the reward is shown.

is the most commonly adopted OS activity by researchers [22–24]. However, engagement with other OS activities, such as data sharing and FAIRification and preregistration of study designs, varies and needs greater support in policies, institutional frameworks and awareness-raising efforts (e.g., awareness of FAIR principles versus being involved in some steps of FAIRification).

Our study also showed that Open or FAIR data management and sharing, an effort-intensive OS activity, stands out as particularly deserving of explicit rewards and credits. Overall, our findings indicated that researchers' OS activities are more driven by various tangible rewards rather than by intangible rewards such as public benefits. In contrast, a previous survey study by Hahnel et al. [11], which explored the rewards that would motivate researchers to share open research data, revealed that the perceived public benefit was considered the second most popular reward, following the citation of research papers. The observed difference may be attributed to variations in survey samples–the Hahnel et al. survey received most responses from India and China, whereas our survey primarily drew responses from Western cultures, including France and the United States.

According to our study, the reward most favoured by researchers was the inclusion of OS indicators in research evaluation and/or career progression processes as a policy measure, followed by tangible incentives of specific funding for OS activities. It is important to note that OS indicators in research evaluation and/or career progression processes call for careful consideration in their application. An approach predominantly centred on qualitative assessments, supplemented by some quantitative measures, is favoured. This aligns with the findings from prior research [4], which highlighted that researchers most highly value 'generating high-quality publications, as assessed by independent qualitative assessment (e.g., peer review)' as a criterion for academic promotion decisions. Furthermore, the recently published UNESCO 'Open Science Outlook' [25] suggests that quantitative indicators for monitoring OS activities' progress and status should be balanced with qualitative proxies.

A main limitation of this study is that our survey sample is quite moderate and not representative of any specific defined population, rather, it consists of those we were able to reach through the authors' networks using the snowball non-probability sampling method. Even though the survey was distributed to various research communities without the specific aim of targeting exclusively researchers engaged in OS, the final set of respondents were primarily individuals familiar with or active in OS, introducing a bias that may reflect the limited uptake of OS activities among many scholars. While the survey data collected through the snowball sampling approach was valuable for informing our later recommendations for OS rewards and incentives endorsed by the RDA community [26], it made the applicability of statistical cross-tabulation analyses irrelevant to this study. As an additional limitation, the seven types of OS activities proposed in the survey are not exhaustive. Even though respondents had the possibility to indicate other OS activities in an open-ended question, these activities were not rated to determine to what extent they should be credited/rewarded.

Despite the survey sample not being representative, as a cross-cultural group of authors, we were able to crosswalk diverse perspectives on what rewards and incentives can mean in different countries and cultures. Thus, the findings of our study can contribute to OS theory and practice by suggesting OS rewards terminology and can inform a broad range of stakeholders, e.g., involved in research and innovation systems, in implementing OS rewarding schemes. In doing so, our work paves the way for broader, more robust quantitative studies in the future.

## Supporting information

**S1 File. Questionnaire used to conduct the survey.**
(PDF)

**S2 File. Full list of organisations used for the dissemination of survey questionnaire.**
(PDF)

**S3 File. PLOS checklist on inclusivity in global research.**
(DOCX)

**S4 File. Aggregated survey responses.**
(CSV)

## Acknowledgments

We thank the respondents who participated in this survey study.

## Author Contributions

**Conceptualization:** Florencia Grattarola, Hanna Shmagun, Christopher Erdmann, Anne Cambon-Thomsen, Mogens Thomsen, Laurence Mabile.

**Data curation:** Florencia Grattarola, Hanna Shmagun.

**Formal analysis:** Florencia Grattarola.

**Funding acquisition:** Jaesoo Kim, Laurence Mabile.

**Methodology:** Florencia Grattarola, Hanna Shmagun.

**Project administration:** Laurence Mabile.

**Supervision:** Laurence Mabile.

**Visualization:** Florencia Grattarola.

**Writing – original draft:** Florencia Grattarola, Hanna Shmagun.

**Writing – review & editing:** Florencia Grattarola, Hanna Shmagun, Christopher Erdmann, Anne Cambon-Thomsen, Mogens Thomsen, Jaesoo Kim, Laurence Mabile.

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
