## [Decision Letter · Decision Letter 0]

16 Apr 2024

PONE-D-24-06329Gaps between Open Science policies and actual recognition systems: Insights from an international surveyPLOS ONE

Dear Dr. Shmagun,

Thank you for submitting your manuscript to PLOS ONE. After careful consideration, we feel that it has merit but does not fully meet PLOS ONE’s publication criteria as it currently stands. Therefore, we invite you to submit a revised version of the manuscript that addresses the points raised during the review process.

We look forward to receiving your revised manuscript.

Kind regards,

Erum Shaikh

Academic Editor

PLOS ONE

Journal Requirements:

"This research received funding from the PARSEC Belmont Forum Collaborative Research Action on Science-Driven e-Infrastructures Innovation (SEI2018). Additionally, FG was funded by the European Union (ERC, BEAST, 101044740). HS received funding from the Korea Institute of Science and Technology Information (No. K-24-L01-C05-S01). ACT received funding from the European Union’s Horizon 2020 research and innovation programme EOSC Future under grant agreement no 101017536 as domain ambassador. "

"This research received funding from the PARSEC Belmont Forum Collaborative Research Action on Science-Driven e-Infrastructures Innovation (SEI2018). Additionally, FG was funded by the European Union (ERC, BEAST, 101044740). HS received funding from the Korea Institute of Science and Technology Information (No. K-24-L01-C05-S01). ACT received funding from the European Union’s Horizon 2020 research and innovation programme EOSC Future under grant agreement no 101017536 as domain ambassador. "

"This research received funding from the PARSEC Belmont Forum Collaborative Research Action on Science-Driven e-Infrastructures Innovation (SEI2018). Additionally, FG was funded by the European Union (ERC, BEAST, 101044740). HS received funding from the Korea Institute of Science and Technology Information (No. K-24-L01-C05-S01). ACT received funding from the European Union’s Horizon 2020 research and innovation programme EOSC Future under grant agreement no 101017536 as domain ambassador. "

Additional Editor Comments:

Reviewer 1

Comments to the Author

1. Is the manuscript technically sound, and do the data support the conclusions?

The manuscript must describe a technically sound piece of scientific research with data that supports the conclusions. Experiments must have been conducted rigorously, with appropriate controls, replication, and sample sizes. The conclusions must be drawn appropriately based on the data presented. Partly

2. Has the statistical analysis been performed appropriately and rigorously? No

3. Have the authors made all data underlying the findings in their manuscript fully available?

The PLOS Data policy requires authors to make all data underlying the findings described in their manuscript fully available without restriction, with rare exception (please refer to the Data Availability Statement in the manuscript PDF file). The data should be provided as part of the manuscript or its supporting information, or deposited to a public repository. For example, in addition to summary statistics, the data points behind means, medians and variance measures should be available. If there are restrictions on publicly sharing data—e.g. participant privacy or use of data from a third party—those must be specified. Yes

4. Is the manuscript presented in an intelligible fashion and written in standard English?

PLOS ONE does not copyedit accepted manuscripts, so the language in submitted articles must be clear, correct, and unambiguous. Any typographical or grammatical errors should be corrected at revision, so please note any specific errors here. Yes

5. Review Comments to the Author

Please use the space provided to explain your answers to the questions above. You may also include additional comments for the author, including concerns about dual publication, research ethics, or publication ethics. (Please upload your review as an attachment if it exceeds 20,000 characters) As the authors state, the responses to this survey are not representative of a specific population, the lack of information on how many people it reaches makes it impossible to calculate the response rate, and according to the data provided and the institutions that participated, the results may be biased by the procedures and opinions of a population that is familiar with and participates in the OS. Apart from this, in material and methods I do not see that the survey was previously tested to see the degree of difficulty and comprehension of the questions and answers. Nor do they mention whether the reliability of the survey questions was assessed, e.g. with the application of Cronbach's alpha (a high Cronbach's alpha values indicate that response values for each participant across a set of questions are consistent). Regarding the open-ended responses, according to the results, there was not much participation, which limits their analysis, and they are barely mentioned in the text . On the other hand, the fact that the survey was sent in several languages may make it difficult to interpret due to cultural and academic environment issues.

I recommend this article, not cited in his paper, which analyzes similar data but with more extension "Habits and perceptions regarding open science by researchers from Spanish institutions" https://doi.org/10.1371/journal.pone.0288313.

6. PLOS authors have the option to publish the peer review history of their article (what does this mean?). If published, this will include your full peer review and any attached files.

Do you want your identity to be public for this peer review? For information about this choice, including consent withdrawal, please see our Privacy Policy. Yes: Remedios Melero

Confidential to Editor

1. Do you have any potential or perceived competing interests that may influence your review? Please review our Competing Interests policy and declare any potential interests that you feel the Editor should be aware of when considering your review. If you have no competing interests, please write "I have no competing interests." About the paper is well written and any ethical question considered regarding the survey but it does not add new methodology, limited conclusions, and I think responses are biased by the type and experience of respondents regarding open science.

2. Did you receive any assistance in preparing this review (e.g. from a post-doc or graduate student)? If yes, please include their name below. No

3. If accepted, do you think this submission should be highlighted on the PLOS ONE website? PLOS ONE does not evaluate manuscripts based on perceived significance or readership. We aim to provide tools for readers to filter and evaluate our publications. (optional) No

Do you want to get recognition for this review on a Web of Science researcher profile?

If you opt in, your Web of Science profile will automatically be updated to show a verified record of this review in full compliance with the journal’s review policy. If you don’t have a Web of Science profile, you will be prompted to create a free account.

No

Reviewer 2Comments to the Author

1. Is the manuscript technically sound, and do the data support the conclusions?

The manuscript must describe a technically sound piece of scientific research with data that supports the conclusions. Experiments must have been conducted rigorously, with appropriate controls, replication, and sample sizes. The conclusions must be drawn appropriately based on the data presented. Yes

2. Has the statistical analysis been performed appropriately and rigorously? Yes

3. Have the authors made all data underlying the findings in their manuscript fully available?

The PLOS Data policy requires authors to make all data underlying the findings described in their manuscript fully available without restriction, with rare exception (please refer to the Data Availability Statement in the manuscript PDF file). The data should be provided as part of the manuscript or its supporting information, or deposited to a public repository. For example, in addition to summary statistics, the data points behind means, medians and variance measures should be available. If there are restrictions on publicly sharing data—e.g. participant privacy or use of data from a third party—those must be specified. Yes

4. Is the manuscript presented in an intelligible fashion and written in standard English?

PLOS ONE does not copyedit accepted manuscripts, so the language in submitted articles must be clear, correct, and unambiguous. Any typographical or grammatical errors should be corrected at revision, so please note any specific errors here. Yes

5. Review Comments to the Author

Please use the space provided to explain your answers to the questions above. You may also include additional comments for the author, including concerns about dual publication, research ethics, or publication ethics. (Please upload your review as an attachment if it exceeds 20,000 characters) The authors examined gaps in Open Science policies and actual recognition systems via an international survey with participants from five continents and 37 countries, N = 230. The survey consisted of 19 questions and was available in English, Korean, and Spanish, and was distributed across RDA-SHARC members’ networks and the RDA community. I like the general research question, it is timely and interdisciplinary, and certainly of interest to the readers of PLOS ONE. The provision of open data and open code is excellent, the Github repository is clear and very descriptive. The sample size is not very large for a cross-sectional study, but sufficient for the excellent and insightful descriptive presentation of the results. The manuscript is well written and concise.

A minor limitation might be that the questions on OS were not soo extensive, only 7 practices were asked (preprint, open access, preregistration, FAIR data, open peer review, citizen science, virtual labs) – and only these practices could be rated whether they should be more incentivized/rewarded. Maybe this could be briefly mentioned in the manuscript (e.g. under limitations), otherwise the title suggests, at least to me, that a broader range of OS practices was investigated. Further criteria could be open source software, reproducibility (e.g. detailed methodologies), Registered Reports (I think it was not clarified whether it felt under preregistration?), open licensing, open policy, open educational resources, open science advocacy and community building, etc. – although the selected 7 criteria are definitely of high interest and relevant.

Related to the title, I was wondering whether “Open Science policies” is the best name. The methods and results section says that “Open Science activities” or “OS practices” were rated regarding respondents’ familiarity, preferences for OS activities, whether these activities are rewarded and should be rewarded. In my opinion, no "policies" (e.g. institutional) in the narrower sense were examined.

There is a spelling error in the Introduction, it says MERRITT instead of MERRIT (line 52).

6. PLOS authors have the option to publish the peer review history of their article (what does this mean?). If published, this will include your full peer review and any attached files.

Do you want your identity to be public for this peer review? For information about this choice, including consent withdrawal, please see our Privacy Policy. No

Confidential to Editor

1. Do you have any potential or perceived competing interests that may influence your review? Please review our Competing Interests policy and declare any potential interests that you feel the Editor should be aware of when considering your review. If you have no competing interests, please write "I have no competing interests." I have no competing interests.

2. Did you receive any assistance in preparing this review (e.g. from a post-doc or graduate student)? If yes, please include their name below. 0

3. If accepted, do you think this submission should be highlighted on the PLOS ONE website? PLOS ONE does not evaluate manuscripts based on perceived significance or readership. We aim to provide tools for readers to filter and evaluate our publications. (optional) Yes, on the homepage

Do you want to get recognition for this review on a Web of Science researcher profile?

If you opt in, your Web of Science profile will automatically be updated to show a verified record of this review in full compliance with the journal’s review policy. If you don’t have a Web of Science profile, you will be prompted to create a free account.

Yes

Reviewers' comments:

Reviewer's Responses to Questions

**Comments to the Author**

1. Is the manuscript technically sound, and do the data support the conclusions?

Reviewer #1: Partly

Reviewer #2: Yes

2. Has the statistical analysis been performed appropriately and rigorously? 

Reviewer #1: No

Reviewer #2: Yes

3. Have the authors made all data underlying the findings in their manuscript fully available?

Reviewer #1: Yes

Reviewer #2: Yes

4. Is the manuscript presented in an intelligible fashion and written in standard English?

Reviewer #1: Yes

Reviewer #2: Yes

5. Review Comments to the Author

Reviewer #1: As the authors state, the responses to this survey are not representative of a specific population, the lack of information on how many people it reaches makes it impossible to calculate the response rate, and according to the data provided and the institutions that participated, the results may be biased by the procedures and opinions of a population that is familiar with and participates in the OS. Apart from this, in material and methods I do not see that the survey was previously tested to see the degree of difficulty and comprehension of the questions and answers. Nor do they mention whether the reliability of the survey questions was assessed, e.g. with the application of Cronbach's alpha (a high Cronbach's alpha values indicate that response values for each participant across a set of questions are consistent). Regarding the open-ended responses, according to the results, there was not much participation, which limits their analysis, and they are barely mentioned in the text . On the other hand, the fact that the survey was sent in several languages may make it difficult to interpret due to cultural and academic environment issues.

I recommend this article, not cited in his paper, which analyzes similar data but with more extension "Habits and perceptions regarding open science by researchers from Spanish institutions" https://doi.org/10.1371/journal.pone.0288313.

Reviewer #2: The authors examined gaps in Open Science policies and actual recognition systems via an international survey with participants from five continents and 37 countries, N = 230. The survey consisted of 19 questions and was available in English, Korean, and Spanish, and was distributed across RDA-SHARC members’ networks and the RDA community. I like the general research question, it is timely and interdisciplinary, and certainly of interest to the readers of PLOS ONE. The provision of open data and open code is excellent, the Github repository is clear and very descriptive. The sample size is not very large for a cross-sectional study, but sufficient for the excellent and insightful descriptive presentation of the results. The manuscript is well written and concise.

A minor limitation might be that the questions on OS were not soo extensive, only 7 practices were asked (preprint, open access, preregistration, FAIR data, open peer review, citizen science, virtual labs) – and only these practices could be rated whether they should be more incentivized/rewarded. Maybe this could be briefly mentioned in the manuscript (e.g. under limitations), otherwise the title suggests, at least to me, that a broader range of OS practices was investigated. Further criteria could be open source software, reproducibility (e.g. detailed methodologies), Registered Reports (I think it was not clarified whether it felt under preregistration?), open licensing, open policy, open educational resources, open science advocacy and community building, etc. – although the selected 7 criteria are definitely of high interest and relevant.

Related to the title, I was wondering whether “Open Science policies” is the best name. The methods and results section says that “Open Science activities” or “OS practices” were rated regarding respondents’ familiarity, preferences for OS activities, whether these activities are rewarded and should be rewarded. In my opinion, no "policies" (e.g. institutional) in the narrower sense were examined.

There is a spelling error in the Introduction, it says MERRITT instead of MERRIT (line 52).

6. PLOS authors have the option to publish the peer review history of their article (what does this mean?). If published, this will include your full peer review and any attached files.

Reviewer #1: **Yes: **Remedios Melero

Reviewer #2: No

---

## [Author Response · Author response to Decision Letter 0]

5 May 2024

We addressed all issues raised. Please see the uploaded "Rebuttal Letter_response to reviewers' comments", where responses to all comments one by one are outlined in detail.

---

## [Decision Letter · Decision Letter 1]

17 Jul 2024

PONE-D-24-06329R1Gaps between Open Science activities and actual recognition systems: Insights from an international surveyPLOS ONE

Dear Dr. Shmagun,

Thank you for submitting your manuscript to PLOS ONE. After careful consideration, we feel that it has merit but does not fully meet PLOS ONE’s publication criteria as it currently stands. Therefore, we invite you to submit a revised version of the manuscript that addresses the points raised during the review process.

I will preface this report by saying that I assumed the role of academic editor of the article after the first phase of review. I reviewed the first-phase reports and the authors' response. I then requested the opinion of a third reviewer, also in light of the fact that one of the two first-level reviewers declined.

I have read the article carefully and believe that it has many weaknesses that need to be carefully addressed by the authors before resubmitting the article to PLOSONE.

There is a first problem related to the power of the sample: the snowballing technique used is not clarified (how large is the initial group? How was the length of the survey decided? Why are 234 responses considered sufficient?).

Obviously, given the low sample, data about single countries are simply unreliable. The abstract omits to disclose the tiny number of respondents and presents the paper as containing significant data for single countries.

The discussion of the bias introduced by selecting people in an RDA group devoted to work on “sharing rewards and credits” is completely inadequate. Moreover, the questionnaire assumes that OS/OA activities should be rewarded and avoids the upstream question of desirability of incentives for OS/OA. In my opinion, the survey cannot be considered as representing opinion of scholars, but only of a very small group of people already working on OS/OA rewarding. From this point of view, I think that the context and limitation of the survey should be more carefully highlighted in the text.

Table 3 omits the number of respondents with  “no opinion”.

The way Figure 4 is constructed is completely incomprehensible. It is derived from an open responses to an open questions. So it is impossible to understand the significance of the percentage. How many indications did each response contains on average? How many indications were derived from each questionnaire? If more than one indication is drawn from each questionnaire, how the percentage should be interpreted?

Moreover, the question of the survey indicated some examples of possible responses that are precisely those we later find in the table, which makes one suspect a bias induced by the way the questionnaire was administered.

The classification of disciplines is completely idiosyncratic (what are applied science or formal science?), adopting no established classification.

For all these reasons, regrettably, I believe that the paper cannot be accepted for publication in its current form. I suggest to the authors to resubmit the article only if they are able to address all the above concerns.

We look forward to receiving your revised manuscript.

Kind regards,

Alberto Baccini, Ph.D.

Academic Editor

PLOS ONE

Reviewers' comments:

Reviewer's Responses to Questions

**Comments to the Author**

1. If the authors have adequately addressed your comments raised in a previous round of review and you feel that this manuscript is now acceptable for publication, you may indicate that here to bypass the “Comments to the Author” section, enter your conflict of interest statement in the “Confidential to Editor” section, and submit your "Accept" recommendation.

Reviewer #1: All comments have been addressed

Reviewer #3: (No Response)

2. Is the manuscript technically sound, and do the data support the conclusions?

Reviewer #1: Yes

Reviewer #3: Yes

3. Has the statistical analysis been performed appropriately and rigorously? 

Reviewer #1: Yes

Reviewer #3: I Don't Know

4. Have the authors made all data underlying the findings in their manuscript fully available?

Reviewer #1: Yes

Reviewer #3: Yes

5. Is the manuscript presented in an intelligible fashion and written in standard English?

Reviewer #1: No

Reviewer #3: Yes

6. Review Comments to the Author

Reviewer #1: (No Response)

Reviewer #3: The study aims at identifying perceptions and expectations of various research communities regarding the recognition and reward of OS practices. In oder to do so, a survey was conducted collecting 230 answers from participants from 37 countries.

The study confirmed something which is well known within the OS community, i.e. that Open Access and FAIR data management are the most commonly adopted OS activities by researchers, and that, as such, are perceived to deserve explicit recognition.

However, a problematic element pointed out by the reviewers remains.

In general, the significance of the sample is not made explicit: 230 responses out of the entire potential population of researchers are not significant. This is a major weakness of the proposed study. I suggest to explain what is the added value of this study more in detail.

Furthermore, in the distribution of responses between countries (p. 6), the percentages of responses for France, US, South Korea, Uruguay and then "other countries" are mentioned. On page 9, it is then stated that positive responses regarding the existence of reward systems came from respondents from France, US, Netherlands, Finland, Slovenia, Spain, UK and Germany, while we have no idea how many responded from most of the countries mentioned here, making the figure even less significant. Moreover, it is not clear whether the respondents came from the same institutions or not. As they were asked to explicit their institutional policies, this information could be relevant.

In addition, the following 2 sentences in the abstract are in contradiction:

"Despite most participants reporting that their organisation had one form or another of formal Open Science policies, the majority indicated that their organisation lacks any initiative or tool that provides specific credits or rewards for Open Science activities. For instance, researchers from France, the United States, the Netherlands and Finland affirmed having such mechanisms in place."

I suggest to reformulate them to make your statement more clear.

7. PLOS authors have the option to publish the peer review history of their article (what does this mean?). If published, this will include your full peer review and any attached files.

Reviewer #1: No

Reviewer #3: **Yes: **Francesca Di Donato

---

## [Author Response · Author response to Decision Letter 1]

13 Oct 2024

The comments and suggestions offered by the editor and the referees have been addressed and indeed helped improve the manuscript. The main concerns we have addressed are: 1) clarification of the snowball non-probability sampling method used and the limitations of a moderate, non-representative sample obtained through it, 2) explanation of the study's added value and why it is worth publishing despite the aforementioned limitations, 3) improvements to figures, such as Fig 1 (distribution of the respondents’ disciplines), Fig 3 (likert scale of proposed OS activities to get rewarded), and Fig 4 (preferences on types of rewards). In addition, we have taken into account all additional suggestions, addressed them, and we now provide a full major revision of our manuscript. Detailed responses to all comments, one by one, are included in the submitted file 'Response to reviewers' comments'.

---

## [Decision Letter · Decision Letter 2]

28 Nov 2024

Gaps between Open Science activities and actual recognition systems: Insights from an international survey

PONE-D-24-06329R2

Dear Dr. Shmagun,

We’re pleased to inform you that your manuscript has been judged scientifically suitable for publication and will be formally accepted for publication once it meets all outstanding technical requirements.

The reviewer's comment indicating major revision reiterates an issue that I raised as academic editor in the previous rounds and that the authors have in this version largely discussed. At this point the issue of the small sample size of the respondents is discussed and the limitations of the paper are clearly stated by the authors. The paper can be published without any further modification by the authors.

Kind regards,

Alberto Baccini, Ph.D.

Academic Editor

PLOS ONE

Additional Editor Comments (optional):

Reviewers' comments:

Reviewer's Responses to Questions

**Comments to the Author**

1. If the authors have adequately addressed your comments raised in a previous round of review and you feel that this manuscript is now acceptable for publication, you may indicate that here to bypass the “Comments to the Author” section, enter your conflict of interest statement in the “Confidential to Editor” section, and submit your "Accept" recommendation.

Reviewer #1: All comments have been addressed

2. Is the manuscript technically sound, and do the data support the conclusions?

Reviewer #1: Partly

3. Has the statistical analysis been performed appropriately and rigorously? 

Reviewer #1: Yes

4. Have the authors made all data underlying the findings in their manuscript fully available?

Reviewer #1: Yes

5. Is the manuscript presented in an intelligible fashion and written in standard English?

Reviewer #1: No

6. Review Comments to the Author

Reviewer #1: I have missed the number of responses, with only the percentage cannot be seen the relevance of participation.

The open ended answers have not be clearly explained/analysed, they could be provided anonymised

7. PLOS authors have the option to publish the peer review history of their article (what does this mean?). If published, this will include your full peer review and any attached files.

Reviewer #1: No

---

## [Editor Report · Acceptance letter]

4 Dec 2024

PONE-D-24-06329R2 

PLOS ONE

Dear Dr. Shmagun, 

I'm pleased to inform you that your manuscript has been deemed suitable for publication in PLOS ONE. Congratulations! Your manuscript is now being handed over to our production team.

Kind regards, 

on behalf of

Prof. Alberto Baccini 

Academic Editor

PLOS ONE